# Old and New Calcineurin Inhibitors in Lupus Nephritis

**DOI:** 10.3390/jcm10214832

**Published:** 2021-10-21

**Authors:** Claudio Ponticelli, Francesco Reggiani, Gabriella Moroni

**Affiliations:** 1Nephrology Division, Ospedale Maggiore Policlinico, 20122 Milan, Italy; 2Nephrology and Dialysis Division, IRCCS Humanitas Research Hospital, via Manzoni 56, 20089 Milan, Italy; francesco.reggiani@humanitas.it (F.R.); gabriella.moroni56@gmail.com (G.M.)

**Keywords:** lupus nephritis, proteinuria, calcineurin inhibitors, cyclosporine, tacrolimus, voclosporin

## Abstract

Calcineurin inhibitors (CNIs) are drugs that inhibit calcineurin, a key phosphatase that dephosphorylates a transcription factor called the nuclear factor of activated T cells (NFAT), allowing its translocation into the nucleus of quiescent T cells. In the nucleus, NFAT activates interleukin 2, which stimulates the proliferation and differentiation of T-cells. CNIs can also stabilize the actin cytoskeleton of podocytes reducing proteinuria. Thanks to these characteristics, CNIs have been often used in the treatment of autoimmune diseases. However, the therapeutic index of CNIs is narrow, and their interactions with other drugs can increase toxicity or reduce efficacy. In lupus nephritis, cyclosporine and tacrolimus have been used both in induction and maintenance therapies. Observational studies and randomized controlled trials showed that both cyclosporine and tacrolimus can increase efficacy. Tolerance is satisfactory if low doses are used and the patient is carefully monitored. More recently, a new CNI, called voclosporin (VCS), has been approved by the Food and Drug Administration for use in lupus nephritis. VCS offers potential advantages over other CNIs. In two large multiethnic trials, VCS was not associated with adverse renal and metabolic events and obtained positive results despite a novel and rapid corticosteroid tapering regime.

## 1. Introduction

Kidney disease is a severe and frequent complication of systemic lupus erythematosus (SLE). Histologically, six classes of lupus nephritis have been identified [1]. The three most frequent and dangerous classes are class III (focal proliferative glomerulonephritis), class IV (diffuse proliferative glomerulonephritis), and class V (membranous glomerulonephritis, often associated with class III or class IV). Immunosuppressive therapy is used to treat these three histological classes, whereas it is not usually used to treat minimal mesangial (class I), mesangial proliferative (class II), or advanced sclerosing (class VI) lupus nephritis. The management of proliferative lupus nephritis usually rests on an initial phase, in which treatment should be aimed to prevent the development of irreversible lesions, followed by a long-term maintenance phase, aimed to keep lupus activity under control. However, despite great progress, lupus nephritis still has an increased risk of end-stage renal disease (ESRD) and mortality [2,3]. Glucocorticoids and cyclophosphamide, either orally or intravenously, are the agents more often used in proliferative lupus nephritis. Both drugs proved to be very effective but can expose patients to serious and even life-threatening complications. Several alternative drugs have been proposed in recent years, including salts of mycophenolic acid, rituximab, and calcineurin inhibitors (CNIs). We will review the use of CNIs in lupus nephritis.

## 2. Calcineurin

Calcineurin is a calcium-calmodulin-dependent serine/threonine heterodimeric protein phosphatase, which is composed of a 61-kD calmodulin-binding catalytic subunit, calcineurin A, and a 19-kD calcium-binding regulatory subunit, calcineurin B. Calcineurin modulates cellular responses to calmodulin, a protein that binds Ca^2+^ [4] and participates to several cellular processes and Ca^2+^ dependent signal transduction pathways. The functions of calcineurin include T-cell activation and regulation of cellular processes, muscle and heart functions, apoptosis, cell differentiation, ion homeostasis, and neuronal functions [5,6]. The alteration of calcineurin activity has been implicated in the pathogenesis of diseases such as cardiac hypertrophy, congenital heart disease, and immunological and neurological disorders. Recent advances regarding calcineurin structure include the determination of its three-dimensional structure. In addition, biochemical and spectroscopic studies, including studies of dinuclear metal ion cofactor and metal ion redox chemistry, are beginning to unravel new aspects of the mechanism of phosphate ester hydrolysis [7]. In autoimmune and alloimmune diseases, calcineurin plays a critical role in the proliferation of T cells (Figure 1). Naïve T cells require three different signals to become activated. Signal 1 is given by the engagement of the T-cell receptor with the antigen, which is presented on the surface of the molecules of the major histocompatibility complex on antigen-presenting cells (APC). Signal 2 is given by the engagement of costimulatory molecules on T cells (CD28, CD154) with their respective ligands on APC (CD80, CD86, and CD40). Lack of co-stimulation leads to T cell anergy [8]. Signal 3 is given by cytokines mediating differentiation and expansion. Activated T cells need further signals for proliferation and differentiation. In activated T cells, there is a large influx of Ca^2+^, which subsequently activates calcineurin. Calcineurin dephosphorylates a family of transcription factors called nuclear-factor activating T cells (NF-ATc 1–4), allowing their translocation to the cell nucleus, where NF-ATc dimerizes with another protein induced by a signal from T-cell receptors (NF-ATn) to form a heterodimeric active nuclear factor (NFAT) [9]. This process activates the gene for interleukin-2 (IL-2), ultimately resulting in secretion of different cytokines, including IL-2, a T-cell growth factor essential for the proliferation of T cells and the generation of effector and memory cells [10].

IL-2, together with IL-15, activates the phosphatyil-inositol-3 kinase (PI3-k), the downstream effector of which is a serine-threonine kinase called mammalian target of rapamycin (mTOR). mTOR regulates cell growth and cell proliferation. mTOR is the subunit of two complexes: mTORC1, or raptor complex, and mTORC2, or rictor complex [11]. mTORC1 regulates cell growth and provides the transduction of proliferative signals to T cells [12]. After receiving the signal for proliferation, activated T lymphocytes require the synthesis of nucleotides to proliferate and differentiate into T-cell effectors [13]. Activated T cells may differentiate into Th1, Th2, and Th17 effector subsets, as well as regulatory T cells. The local cytokine environment, established by innate immunity, favors the differentiation into Th1 and Th17 cells [14]. These alloreactive T cells mediate kidney injury through direct contact with tubular epithelial cells and endothelial cells, as well as through the release of cytokines and chemokines that may cause necrosis of the renal tissue. Th1 cells, which release interferon-γ and IL-2, mediate both the cellular arm of the immune system and B-cell class switching to complement immunoglobulin G (IgG) fixing antibodies. Th17 differentiation, which is stimulated by transforming growth factor β and by several cytokines, produces the proinflammatory IL-17. This interleukin plays a key role in the immune response. It mediates inflammation and stimulates production of inflammatory cytokines and inflammatory chemokines that promote the recruitment of neutrophils and macrophages [15].

## 3. Calcineurin Inhibitors

*Mechanisms of action.* Two calcineurin inhibitors (CNIs) have been traditionally used in systemic lupus erythematosus: cyclosporine (CsA) and tacrolimus (TAC). CsA is a cyclical polypeptide composed of 11 amino acids derived from the fungus Tolypocladium inflatum. Tacrolimus is a macrolide derived from the fungus Streptomyces tsukubaensis. These CNIs have a different structure but similar mechanism of action and pharmacological effect. Both inhibit the intracellular activation of calcineurin. The original formulation of CsA has been replaced by a new microemulsion, called Neoral^®^. Compared to the old formulation of CsA, Neoral^®^ has a more rapid and complete absorption. TAC has a new formulation as well, called Advagraf^®^, the pharmacological properties of which allow once-daily administration.

Recently, voclosporin (VCS), a new CNI, was approved by the U.S. Food and Drug Administration for use in the treatment of lupus nephritis. VCS is a semisynthetic structural analogue of CsA, but with a methyl group added to the aminoacid-1 residue. This chemical modification increases the potency of VCS compared to CsA [16]. Moreover, VCS has more stable pharmacokinetic and pharmacodynamic properties and a better metabolic profile (less risk of glucose intolerance and lipid disorders) than CsA [17].

After oral administration and absorption in the small intestine, 60% of CNIs bind to erythrocytes in the blood, 33% bind to lipoproteins, and a minimal fraction circulates freely. All CNIs are metabolized in the liver by the enzymes of cytochrome P-450 (CYP450). This family of enzymes is responsible for around three-quarters of all drug metabolism reactions that occur in human populations. Many isoforms of cytochrome P450 exist, but most reactions are undertaken by CYP2C9, CYP2C19, CYP2D6, CYP3A4, and CYP3A5. Apart from single nucleotide polymorphism, which may result in variant CYP450 enzyme expression and/or activity, factors that inhibit the activity of CYP450 increase the bioavailability of CNIs, whereas CYP450 inducers reduce their bioavailability [18]. The metabolites are mainly excreted with bile and in a small amount with urine. Many factors may influence the bioavailability and pharmacokinetics of CNIs (Table 1).

CNIs are lipophilic drugs and can easily enter the cells where they exert their activity. The intracellular concentration of CNIs is regulated by P-glycoprotein (also called ATP-binding cassette ABC B1 or multi-drug resistance protein). This protein can limit excessive intracellular concentration of toxins and drugs, including CNIs, increasing their blood concentration. Besides possible genetic mutations, several factors can inhibit or increase the activity of P-glycoprotein (Table 2).

The main effect of CNIs consists of inhibiting T-cell functions. CNIs suppress the immune system and act selectively on T cells by slowing down their growth. After entering the circulation, the lipophilic CNIs easily diffuse through the cell membrane. CNIs are prodrugs, and within cells they bind to a specific protein receptor, called cyclophilin for CsA and FK binding protein 12 for TAC, a ubiquitous isomerase that catalyzes a rate-determining isomerization in protein folding in vitro [19]. These receptors mediate the cis/trans isomerization of proline imino bonds, allowing the slow refolding phase in which proteins fold to their native three-dimensional structure, which is necessary for their function as enzymes or structural proteins. The complex CNI-receptor binds to and inhibits calcineurin. Because its dephosphorylation is inhibited by CNIs, NFAT cannot enter the nucleus and cannot encode and synthesize IL-2 and other cytokines produced by T-helper-1 cells. IL-2 is a T-cell growth factor essential for the proliferation of T cells and the generation of effector and memory cells [10]. The inhibited production of IL-2 prevents the proliferation and differentiation of cytotoxic and other effector T cells. In addition, as the secretion of IL-2 by Th1 cells enhances polyclonal IgM by activating peripheral blood cells [20,21] and activates B cells to secrete antibodies, the production of humoral antibodies is reduced. CNIs can also block the activation of Jun N-terminal kinases and p38 signaling pathways triggered by antigen recognition, making the drugs specific inhibitors of T-cell activation. Other important mechanisms of action are represented by the increased expression of transforming growth factor-ß1 and inhibited synthesis of interferon γ, colony-stimulating factor, and macrophage-activating factor, which provide signals activating macrophages and monocytes and play an important role in inflammatory processes. All of these effects are rapid in onset, dose-dependent, and often quickly reversible after the treatment is interrupted [22]. VCS, like CsA, exerts its immunosuppressant effects by inhibition of the calcineurin signal-transduction pathway. In vitro, VCS suppresses diverse immune functions more potently than CsA [23].

Apart from their systemic immunosuppressive activity, CNIs could exert a protective effect on podocytes by different mechanisms. CsA blocks the calcineurin-mediated dephosphorylation of synaptopodin, an actin-associated protein that modulates actin-based shape and motility of renal podocyte foot processes [24]. By stabilizing synaptopodin, CsA can also prevent intrapodocyte transient receptor potential cation channel subfamily c member 6 (TRPC6) localization and activity, which contributes to podocyte dysfunction and proteinuria [25]. The inhibited access of NFAT to the nucleus prevents its binding to the gene promoter encoding uPAR, which activates β3 integrin pathway [26]. The impaired cholesterol extrusion from fatty podocyte in nephrotic syndrome is mediated by NFAT activation, which is inhibited by CsA [27]. Finally, murine podocytes express IL-2R, which upregulates pro-apoptotic molecules and induces podocyte injury. CsA can inhibit activation of IL-2R by blocking the synthesis of IL-2 [28]. Similar mechanisms of action may be exerted by VCS. TAC may also protect podocytes by stabilizing cytoskeleton. In experimental studies of podocyte injury, TAC prevented the nuclear translocation of calcineurin binding protein 1, restored podocyte injury, and inhibited podocyte apoptosis, so reducing proteinuria and attenuating renal damage [29]. In puromycin aminonucleoside-treated mouse podocytes, pre-incubation with CsA and TAC restored the distribution of the actin cytoskeleton, increased the expression of synaptopodin and podocin, improved podocyte viability, and reduced the migrating activities of podocytes [30].

*Side effects* (Table 3). CNIs have a narrow therapeutic index. CNI-related toxicity is usually dose- and time-dependent. However, adverse events may also be caused by interactions with other drugs that can increase unwanted effects or reduce the therapeutic efficacy.

Nephrotoxicity. It is one of the most worrying adverse events of CNIs. It is usually dose-dependent but can also be caused by drug–drug interactions influencing bioavailability and intracellular concentrations. Acute nephrotoxicity is characterized by a reversible decrease in glomerular filtration rate (GFR), but in severe cases it may be associated with histologic lesions of thrombotic microangiopathy. Chronic nephrotoxicity is characterized by nonspecific arteriolar lesions, patchy interstitial fibrosis, tubular atrophy, and focal and segmental and global glomerular sclerosis, which can be progressive and lead to irreversible lesions (Figure 2). Chronic nephrotoxicity has been considered for a long time as the Achilles’ heel of CNIs [31]. However, chronic changes are often caused by previously unrecognized immunologic injuries [13]. The differential diagnosis is difficult even for an expert pathologist. Unfortunately, CNI blood levels are of little help and may even be misleading because they do not correlate with the intracellular concentrations [32].

Arterial hypertension. It is caused by the vasoconstrictive effect of CNIs on afferent pre-glomerular arterioles, which leads to decreased GFR and natriuresis and subsequent salt and water retention [33]. This effect is amplified by an epithelial action of CNIs, which can directly activate the thiazide-sensitive Na-Cl cotransporter of the distal convoluted tubule [34]. The clinical impression is that arterial hypertension is less pronounced with TAC and VCS than with CsA.

Diabetes mellitus. CNIs can induce glucose intolerance by several mechanisms, including a decrease in insulin secretion [35] and an increase in insulin resistance [36]. There is general agreement that TAC is more diabetogenic than CsA [37], probably because TAC potentiates glucolipotoxicity in β cells [38]. VCS does not reduce insulin secretion and causes diabetes less frequently than TAC [39].

Dyslipidemia. It is frequent with CsA but uncommon with TAC and VCS. CsA may increase LDL and VLDL cholesterol levels, VLDL triglyceride levels, as well as apolipoprotein B and lipoprotein(a), while reducing HDL concentration. CsA induces dyslipidemia by downregulating the expression of the LDL receptor that mediates lipoprotein clearance [40]. CsA can also reduce lipolysis by inhibiting calcineurin, which is able to activate lipolysis [41], and can increase genes and/or proteins involved in hepatic lipogenesis [42], and may decrease the transport of cholesterol to the intestines by inhibiting 26-hydroxylase, an enzyme involved in the formation of bile acids from cholesterol. On the other hand, dyslipidemia may diminish the expression and functional activity of P-glycoprotein, leading to increased CsA exposure [43].

Hyperuricemia. CsA may cause hyperuricemia more often than TAC or VCS [17]. The specific mechanism for CNI-induced hyperuricemia is unknown, but it probably involves alterations in tubular transport of uric acid. Hyperuricemia may lead to the development of severe acute gouty arthritis and chronic tophaceous gout. There is increasing evidence suggesting a role of hyperuricemia in favoring arterial hypertension and progression of chronic kidney disease [44].

Neurologic complications. Tremor, burning paresthesia, headache, flushing, depression, confusion, and insomnia are dose-dependent and are more frequent and more severe with TAC compared to CsA. Convulsions, aphasia, paralysis, and disabling pain syndrome can also occur [45]. Other rare complications are hearing loss, tinnitus, or otalgia.

Dermatologic complications. These side effects include hypertrichosis and gingival hyperplasia with CsA, and alopecia with TAC.

Alimentary tract complications. Gastric discomfort and diarrhea are more frequent with TAC but are rarely severe.

Electrolyte disorders. Hypomagnesemia and hyperkalemia are the more frequent electrolyte abnormalities associated with CNIs. Hypomagnesemia is caused by a decreased renal reabsorption of magnesium and chronic renal magnesium wasting [46]. Hypomagnesemia may be responsible for mild hypocalcemia, hyperexcitability, fatigue, and tachycardia. Hyperkalemia is usually mild. It can be due to a reduced expression of the Na-K-2Cl-cotransporter at the apical membrane of tubular epithelial cells caused by inhibition of calcineurin [47].

## 4. Old CNIs in Lupus Nephritis

The ability to inhibit T-cell function and to reduce proteinuria through non-immunologic mechanisms, and their safety in pregnancy and lactation, have made CNIs an attractive therapeutic option in lupus nephritis. Several studies and meta-analyses investigated the role of CsA and TAC in the treatment of lupus nephritis, either with corticosteroids alone or as a component of multitarget therapy, and both for initial and maintenance therapy [48].

Cyclosporine. CsA started to be used in lupus nephritis in 1989. A few observational studies reported improvement in proteinuria and stable kidney function in lupus patients given CsA together with corticosteroids [49,50,51,52]. Four randomized controlled trials (RCTs) also confirmed the efficacy of CsA. In one study, 40 children with active proliferative lupus nephritis were randomized to receive CsA Neoral^®^ (5 mg/kg/d) alone or prednisolone (2 mg/kg/d) and cyclophosphamide (CYC) (2 mg/kg/d) for 1 year. Proteinuria significantly decreased in both groups. Creatinine clearance slightly declined, while growth velocity significantly improved in the CsA group [53]. In another RCT, 42 adults with lupus membranous nephropathy were randomized to alternate-day prednisone alone or in combination with low-dose CsA for 1 year or with intermittent intravenous CYC (IV CYC) for six doses. At 1 year, the cumulative probability of remission was 27% with prednisone, 60% with IV CYC, and 83% with CsA. Adverse effects during the 12-month protocol included diabetes (one with prednisone and two with CsA), pneumonia (one with prednisone and two with CsA), and localized herpes zoster (two with IV CYC). Frequent relapses occurred after CsA withdrawal [54]. Zavada et al. in the Cyclofa-Lune study randomly assigned 40 patients with active proliferative lupus nephritis to CYC or CsA. They found that at the end of the initial therapy, 24% of patients treated with CYC achieved remission and 52% response, compared with 26% and 43% of patients treated with CsA; at the end of the maintenance phase, 14% of patients treated with CYC and 37% with CsA had remission, and 38% and 58% respectively responded. Treatment with CsA was associated with transient increase in blood pressure and reversible decrease in GFR [55]. A fourth RCT in 75 patients with class IV lupus nephritis tested the efficacy of CsA as maintenance therapy. Patients received as initial treatment three intravenous pulses of methylprednisolone followed by prednisone in tapering doses and oral CYC (1–2 mg/kg/day). After 3 months, CYC was stopped, and patients were randomized to receive either CsA (mean dose 2.1 mg/kg/d) or azathioprine (mean dose of 0.9 mg/kg/d). Treatment continued for 4 years. The primary end point was the occurrence of lupus flares. Seven lupus flares occurred in the CsA group and eight in the azathioprine group. At the last follow-up, proteinuria was significantly decreased in both groups; however, a higher percentage of patients treated with CsA (42% vs. 15%) had undetectable proteinuria. Creatinine clearance and blood pressure levels did not change significantly in either group. At repeated kidney biopsy, the activity index decreased significantly in both groups, and the chronicity index slightly increased without any difference between CsA and azathioprine. Among the side effects, leukopenia and infections were more frequent in the azathioprine group, and arthralgias and gastrointestinal disorders in the CsA group [56].

Other observational studies confirmed that CsA can reduce proteinuria both in proliferative and membranous lupus nephritis [57,58]. Argolini et al. [59] compared the efficacy of CsA, mycophenolate mofetil (MMF), and azathioprine as maintenance treatment in 106 patients with lupus nephritis. No difference was found among treatments in maintaining stable kidney function and complete remission. After 8 years of follow-up, the CsA-treated group showed a trend towards reduced frequency of renal flares compared to the groups treated with MMF or Azathioprine. No difference in the number and severity of side effects was found among groups. These data suggest that using CsA in the long-term maintenance treatment of lupus nephritis may be effective and safe.

Compared to the first studies that used high doses of CsA, the most recent studies tend to use low-dose CsA Neoral^®^, starting with doses of 4 mg/kg/d that are gradually lowered to a maintenance does of 2 mg/kg/d or even lower. This policy allowed the prevention of hypertension and kidney toxicity and long-term use of CsA. However, the serologic activity of lupus did not burn out in all patients. This dissociation between clinical and serological activity is not completely unexpected because CsA mainly inhibits proinflammatory cytokines and T-helper activity, while it has only an indirect effect on humoral activity [60].

Tacrolimus. TAC started to be used in lupus nephritis 20 years after CsA. The first studies were performed in Asian countries [61,62,63,64,65]. TAC alone or in combination with other immunosuppressive drugs was tested in different forms of lupus nephritis, either for induction therapy or for treating refractory lupus nephritis. Consistent proteinuria reduction, stable serum creatinine, and little prevalence of side effects were reported. However, the follow-ups were limited to 6 months, and the incidence of renal flare after discontinuation of therapy was higher than observed with other induction agents. After these preliminary experiences, several observational trials and RCTs were performed. A meta-analysis of five RCTs including 225 Chinese patients compared efficacy and safety of TAC vs. IV CYC in the induction treatment for lupus nephritis. TAC significantly increased complete remission, total response rate, and serum albumin level; it decreased proteinuria and the systemic lupus erythematosus disease activity index (SLEDAI) compared to CYC. The rates of gastrointestinal symptoms and amenorrhea were significantly lower in the TAC group [66].

A Korean meta-analysis of 9 RCTs including 972 patients compared the efficacy and safety of TAC vs. MMF and IV CYC. TAC showed a significantly higher overall response rate than CYC and was more efficacious than MMF. In terms of safety, TAC showed the highest probability of decreasing the risk of serious infections, followed by MMF and CYC [67].

A new therapeutic approach with a multitarget therapy was evaluated in a large RCT including 26 Chinese centers. Adults with biopsy-proven lupus nephritis were randomized to receive a multitarget therapy (TAC 4 mg/day, and MMF 1 g/day) or IV CYC 0.75 g every 4 weeks. Both groups received 3 days of pulse methylprednisolone (MPP) followed by a tapering course of oral prednisone therapy. After 6 months of therapy, a significantly greater number of patients in the multitarget group (45.9%) than in the CYC group (25.6%) showed complete remission. The overall response incidence was significantly higher in the multitarget group than in the CYC group (83.5% vs. 63.0%), and the median time to overall response was shorter in the multitarget group. Incidence of adverse events (50.3% vs. 52.5%) did not differ between the two groups [68]. As an extension of the prior multitarget therapy trial, an 18-month open label multicenter study that included patients who had responded at 24 weeks was performed. Patients who received multitarget induction therapy continued to receive the same combination of drugs, and patients who had received IV CYC received azathioprine plus prednisone. The multitarget and azathioprine groups had similar cumulative renal relapse rates (5.47% vs. 7.62%). Serum creatinine levels and eGFR remained stable in both groups. The azathioprine group had more adverse events, 44.4% versus 16.4% [69]. Recently, another RCT from Hong Kong reported the 10-year outcome of 150 patients with lupus nephritis treated with MMF or TAC combined with prednisone for induction. At 6 months, complete renal response rate was similar between the MMF (59%) and TAC-treated patients (62%). Responders were switched to azathioprine. After 118.2 ± 42 months, proteinuric and nephritic renal flares occurred respectively in 34% and 37% of the MMF arm, and 53% and 30% of the TAC arm. The cumulative incidence of a composite outcome consisting of reduced eGFR by 30% or more, chronic kidney disease stage 4/5, or death at 10 years was 33% in both groups [70].

CNIs have not been included among recommended treatments in the 2012 KDIGO Clinical Practice Guideline for Glomerulonephritis [71]. However, CNIs are recommended for initial therapy of LN in the more recent EULAR/ERA-EDTA recommendations [72].

## 5. New CNI in Lupus Nephritis

Voclosporin, the new CNI, has been developed and recently approved for the treatment of lupus nephritis. This drug has more stable pharmacokinetic and pharmacodynamic profiles than CsA, making monitoring of blood levels unnecessary [73]. VCS also has a better metabolic profile than CsA, resulting in lower risk of developing diabetes or dyslipidemia [39]. These characteristics prompted investigations into the potential use of VCS in lupus nephritis. AURA-LV was a phase II, multicenter, randomized, double-blind, placebo-controlled trial of two doses of VCS (23.7 mg or 39.5 mg, each twice daily) versus placebo in combination with MMF (2 g/d) and rapidly tapered low-dose prednisone for induction of remission in lupus nephritis. In 20 countries, 265 patients with lupus nephritis were recruited and randomized to treatment for 48 weeks. Complete remission at week 24 was achieved by 29 (32.6%) subjects in the low-dose VCS arm, 24 (27.3%) subjects ithe high-dose VCS arm, and 17 (19.3%) subjects in the placebo group. The significantly greater rates of complete remission in the low-dose and high-dose VCS arms persisted at 48 weeks. There were more serious adverse events in both VCS groups, and more deaths in the low-dose group (11.2%) compared to placebo (1.1%) and high-dose VCS (2.3%) [74]. Recently, the phase III AURORA 1 trial was reported. In this multiethnic study, 357 patients with a biopsy-proven diagnosis of proliferative or membranous lupus nephritis, an eGFR > 45mL/min/1.73 m^2^, and a urinary protein to creatinine ratio (UPCR) ≥ 1.5mg/mg were randomly assigned to oral VCS (23.7 mg twice daily) or placebo; all patients were also given MMF (1g twice daily) and rapidly tapered low-dose oral steroids. At week 52, the primary end point of complete renal response (defined as UPCR ≤ 0.5 mg/mg and eGFR ≥ 60 mL/min/1.73 m^2^) was achieved in 41% of patients in the VCS group and 23% of patients in the placebo group (*p* < 0.0001). Serious adverse events occurred in 21% of patients in VCS and 21% of patients in placebo. One patient randomized to VCS (>1%) and five patients in the placebo group (3%) died. These results were obtained with marked and rapid reduction of standard corticosteroid dosages. Some secondary end points, such as time to achieve UPCR < 0.5 and time to 50% reduction in UPCR, were achieved. At week 52, the VCS arm showed neither a significant decrease in eGFR nor an increase in blood pressure in lipid or glucose profile [75].

In January 2021, based on positive results from the pivotal phase II and III trials, oral voclosporin received its first approval in the USA for use in combination with a background immunosuppressive therapy regimen for adults with active lupus nephritis.

In Table 4 are summarized the main studies regarding the use of three CNIs in LN.

## 6. CNI and Pregnancy

Pregnancy is not contraindicated in LN, but preconception counseling and close monitoring during pregnancy are crucial to prevent the risks of an unplanned pregnancy. The prognosis is good for the mother if glomerular filtration rate is ≥60 mL/min, proteinuria is <1 g/24 h, and blood pressure is under control. However, the presence of renal insufficiency, hypertension, and antiphospholipid antibodies can increase the risk of lupus flares and preeclampsia. Miscarriage, premature delivery, and heart problems are the major complications that can occur in babies from mothers with LN. The Food and Drug Administration classifies CsA as category C; that is, although risk to the fetus has not been ruled out in human and/or animal studies, the benefits of use may exceed the risk. Meta-analysis studies in rheumatic patients did not see any significant difference in birth defects between pregnancies with prenatal exposure to CsA and controls [76]. CsA does not appear to be teratogen, but may increase the risk of low birth weight [77]. Reports of using TAC to treat lupus nephritis in pregnancy are limited. In a small study, mothers taking TAC had healthy newborns and continued breastfeeding [78]. A meta-analysis of 10 studies found TAC-treated transplant-recipient patients had a lower risk of gestational hypertension while CsA-treated patients had lower incidence of caesarean section and a higher incidence of live birth [79].

## 7. Conclusions

In the last years, there has been agreement that some patients with lupus nephritis may need less intense immunosuppression to avoid long-term deleterious side effects. For initial therapy, the 2019 European League Against Rheumatism (EULAR) and European Renal Association–European Dialysis and Transplant Association (ERA–EDTA) guidelines recommend pulse IV methylprednisolone followed by oral prednisone (0.3–0.5 mg/kg/day) plus MMF or low-dose IV CYC (500 mg × 6 biweekly doses) or a combination of glucocorticoids with MMF and a CNI, especially TAC in cases with nephrotic range proteinuria [80]. Thus, CNIs are now considered as an alternative to MMF or CYC for induction therapy of lupus nephritis. However, in patients with a rapidly progressive course, established kidney dysfunction, abundant crescents and/or glomerular capillary necrosis at kidney biopsy, CYC is still preferable to CNIs [79].

We feel that low-dose CNIs are particularly indicated also in the chronic management of lupus patients with proteinuria. TAC may be considered to have the same role and the same limits of CsA in treating lupus nephritis. TAC is more expensive and may cause diabetes and neurologic symptoms more frequently than CsA. However, TAC less frequently causes hypertension and dyslipidemia compared to CsA and does not cause esthetic changes. Recommendations with the use of TAC are comparable to those given for CsA. The initial doses may range around 0.1 mg/kg per day, with gradual reduction to the lowest possible doses. At present, there is a preference for the use of TAC. However, most studies with TAC in lupus nephritis have been conducted in patients from Asia, and the results can not easily be applied to other ethnicities. Indeed, there are important racial/ethnic differences in metabolic phenotypes due to differences in single nucleotide polymorphisms [81]. Another relevant limitation is the fact that the short-term outcome measure of these studies is proteinuria. However, TAC may reduce proteinuria also through nonimmune mechanisms, as described above, and considering reduced proteinuria as a marker of improved disease activity may be misleading. In addition, further data on long-term renal and cardiovascular outcomes and strategies to improve tolerability and safety are required for TAC. To prevent nephrotoxicity and other important complications, CNIs should be avoided in patients with a baseline eGFR < 40 mL/min, and the doses should be reduced if the eGFR increases more than 30% over the baseline. CNI blood levels should not be used as the only decision-making tool to adjust doses, as they do not provide adequate information on intracellular drug concentration and pharmacological activity.

VCS offers potential advantages over other CNIs. The drug was tested in patients of different ethnicities. In the available trials, VCS was not associated with adverse renal and metabolic events and obtained positive results despite a novel and rapid corticosteroid tapering regime. This may open new perspectives on the possibility of replacing more toxic drugs with VCS. However, a long-term study would be necessary to completely exclude nephrotoxicity. In addition, the levels of GFR and SLEDAI scores were similar in patients given VCS or placebo, suggesting that the effects of VCS were not related to its immunosuppressive activity. It seems more probable that, similarly with the old CNIs, VCS can exert an antiproteinuric effect by stabilizing the actin cytoskeleton in kidney podocytes. Finally, a head-to-head comparison between VCS and low-dose CsA or TAC would be needed to confirm the superiority of VCS over the old CNIs [82].

## Figures and Tables

**Figure 1 jcm-10-04832-f001:**
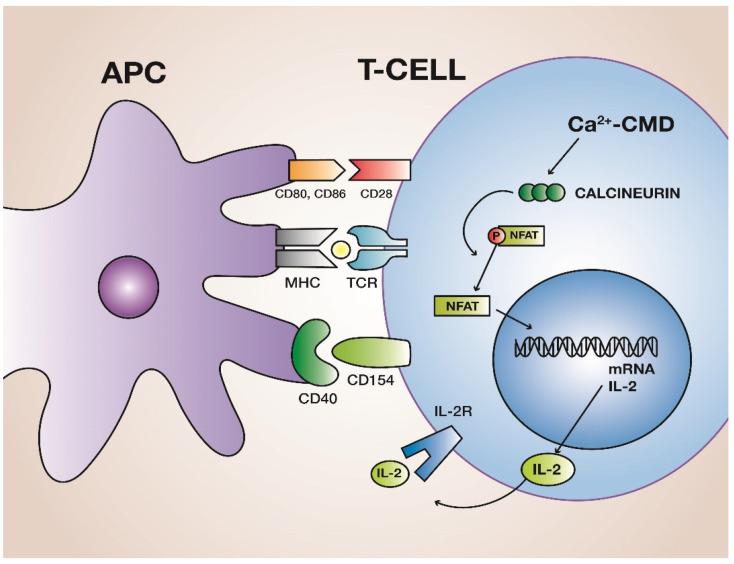
Contact between a peptide of the major histocompatibility complex (MHC) and T-cell receptor (TCR) does not activate naïve T cells. To be activated, T cells also require co-stimulation given by contact between the co-stimulatory molecules of antigen presenting cells (APC), CD80–86, and CD40, and the corresponding molecules on T cell, CD28, and CD 154, respectively. In the activated T cell, there is a large influx of Ca++ ions, which bind to calmodulin (CDM). This complex activates calcineurin, which dephosphorylates NF-AT, allowing its translocation to the nucleus, where NF-AT collaborates in the synthesis of IL-2. IL-2 binds to its receptor (IL-2R), which activates a cascade of kinases, eventually providing the signal for T-cell proliferation and differentiation.

**Figure 2 jcm-10-04832-f002:**
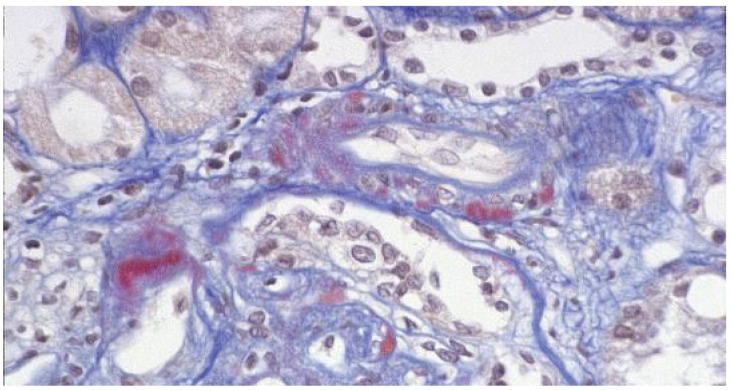
Light microscopy: AFOG stain. Nodular hyaline material is present in the outer media of some arterioles in a patient treated with calcineurin inhibitors.

**Table 1 jcm-10-04832-t001:** Calcineurin inhibitors (CNIs) are metabolized in the liver by the enzymes of cytochrome P-450 (CYP450). CYP450 inhibitors increase the bioavailability of CNIs, while CYP450 inducers decrease it. CCBs, calcium channel blockers.

CYP450 Inhibitors	CYP450 Inducers
CNIs Bioavailability	
*Increased*	*Decreased*
Older age	Childhood
Obesity	Malnutrition
Smoking	Biliary diversion
Liver disease	Diarrhea
Grapefruit	Antiepileptic drugs
Antifungal azoles	Nafcillin
Macrolides	Oxacillin
Quinolones	Rifampicin
Non-dihydropyridine CCBs	Rifabutin

**Table 2 jcm-10-04832-t002:** Several factors can inhibit or increase the activity of P-glycoprotein, which regulates the intracellular concentration of CNIs. This protein can limit an excessive intracellular concentration of toxins and drugs, including CNIs, increasing their blood concentration. CNIs, calcineurin inhibitors.

P-Glycoprotein Inhibitors	P-Glycoprotein Inducers
*Decrease* CNIs Plasma Levels and Increase Intracellular Concentration	Increase CNIs Plasma Levels and Decrease Intracellular Concentration
Macrolides *(erythromycin*, *clarithromycin)*	Antiepileptic Drugs *(carbamazepine*, *phenobarbital*, *phenytoin)*
Proton pump inhibitors *(omeprazole, lansoprazole)*	Glucocorticoids *(dexamethasone)*
Calcium channel blockers *(verapamil*, *diltiazem*, *felodipine*, *nifedipine)*	Rifampicin
Antiarrhythmic drugs *(amiodarone*, *quinidine)*	Tenofovir
Antidepressant drugs *(paroxetine*, *sertraline)*	Antidepressant drugs *(nefazodone*, *trazodone)*
Antifungal drugs *(ketoconazole)*	Alpha-blockers *(prazosin)*
Cyclosporine	Doxorubicin
Fenofibrates	

**Table 3 jcm-10-04832-t003:** Main complications of CNIs.

Chronic nephrotoxicity *Similar for CsA and TAC.*	Dyslipidemia *More frequent with CsA.*
Thrombotic microangiopathy *Dose-dependent. Similar for CsA and TAC.*	Gum hyperplasia *Typical of CsA. Aggravated by* *simultaneous use of CCB.*
Neurological complications *More frequent with TAC.*	Increased hair growth *Typical of CsA. TAC may rarely cause alopecia.*
Diabetes *More frequent with TAC.*	Hearing loss *More frequent with TAC.*
Gastrointestinal troubles *Similar for CsA and TAC.*	Electrolyte disorders *Hypomagnesemia, hyperkalemia.*
Hypertension *More frequent with CsA.*	Hyperuricemia *Similar for CsA and TAC.*

CsA = cyclosporine. TAC = tacrolimus (TAC). CCB = calcium channel blockers.

**Table 4 jcm-10-04832-t004:** Summary of the main studies regarding the use of CNIs in lupus nephritis.

	Author, Year	Study Type	Study Population	Drugs	Follow-Up	Endpoint	Results	Notes
CsA—Induction	Fu [53]	RCT	40 children	CsA Neoral vs. prednisolone + CYC	1 year	Reduction of proteinuria	Reduction of proteinuria with both drugs	Neoral well tolerated, better growth rate.
	Austin [54]	RCT	42 patients (LN class V)	Prednisone vs. prednisone + CsA vs. prednisone + IV CYC	1 year	Remission of proteinuria	27% with prednisone, 60% with IV CYC, and 83% with CsA	Significantly more relapse of nephrotic syndrome after CsA than after IV CYC
	Zavada [55]	RCT	40 patients (active LN)	Regimens based on CYC vs. CsA	End of induction and maintenance phase	Remission and response to therapy	CYC (24% remission, 52% response); CsA (26% remission, 43% response)	Transient increase in blood pressure and reversible decrease in GFR with CsA
	Sheikholesla-mi [57]	OS	27 patients (resistant LN)	CsA added to steroid + MMF or steroid + CYC	40.7 ± 24.9 months	Complete and partial renal remission	66.9% and 25.7%	With CsA stable creatinine, <proteinuria and anti-dsDNA titer, >of C3 and C4
	Sumethkul [58]	OS	62 patients (active LN)	CsA + MMF and prednisolone, or CsA + prednisolone	1 year	Complete and partial renal remission	90.3% and 40.8%	Non-renal activity including arthritis, alopecia, hematologic and cutaneous conditions improved in all patients.
CsA—Maintenance	Moroni [56]	RCT	75 patients (LN class IV)	CsA vs. AZA (both after steroids and CYC)	4 years	Lupus flares	7 lupus flares with CsA, 8 with AZA	No difference in creatinine clearance. In both groups activity index decreased significantly and the chronicity index slightly increased.
	Argolini [59]	OS	106 patients	CsA vs. MMF vs. AZA	8 years	Complete remission	79.4% of CsA vs. 83.3% of MMF and 77.8% of AZA	Flares-free survival curves and incidence of side-effects were not different.
TAC	Miyasaka [62]	RCT	63 patients (persistent LN)	TAC vs. placebo	28 weeks	Change in LNDAI	31% decrease with TAC, vs. 38% increase with placebo	Treatment-related adverse events occurred in 93% with TAC and 80% with placebo.
	Deng [66]	MA	225 Chinese patients—5 RCTs	TAC (oral or IV) vs. IV CYC for induction therapy	\	Complete remission, response rate, serum albumin, anti-dsDNA, proteinuria, SLEDAI	TAC superior in all endpoints	TAC safer than IV CYC
	Lee [67]	MA	972 Korean patients—9 RCTs	TAC vs. MMF vs. IV CYC for induction therapy	\	Overall response rate (complete remission plus partial remission)	TAC showed higher overall response rate than CYC than MMF	Better overall response with MMF. Less serious infections with TAC.
	Liu [68]	RCT	368 patients	TAC + MMF vs. IV CYC (both preceded by pulse MMP)	24 weeks	Complete remission	46% with TAC plus MMF vs. 26% with IV CYC showed complete remission	Higher overall response incidence with TAC plus MMF. Shorter median time to overall response with TAC.
	Zhang [69]	CT	206 patients	TAC + MMF vs. AZA + prednisone	2 years	Renal relapse	Similar cumulative renal relapse rates	More adverse events with AZA.
	Mok [70]	RCT	150 patients	MMF vs. TAC, + high-dose prednisolone	10 years	Complete response	Complete renal response rate similar between MMF (59%) and TAC (62%)	Proteinuric and nephritic renal flares in 34% and 37% of the MMF, and 53% and 30% of the TAC groups.
VCS	Rovin [74]	RCT	265 patients	VCS vs. placebo, with MMF and rapidly tapered low-dose oral corticosteroids	24 weeks	Complete response	33% complete renal response in low-dose VCS group, 27% in high-dose VCS group, 19% in placebo group	More serious adverse events in VCS groups. More deaths in the low-dose group (11%) and high-dose VCS (2%) than placebo (1.1%).

AZA, azathioprine; CsA, cyclosporine; CT, controlled trial; CYC, cyclophosphamide; GFR, glomerular filtration rate; MA, meta-analysis; MMF, mofetil mycophenolate; OS, observational study; RCT, randomized controlled trial; TAC, tacrolimus; VCS, voclosporin. + means “and”.

## Data Availability

Not applicable.

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
