# Peer review of "Old and New Calcineurin Inhibitors in Lupus Nephritis"

_jcm, 2021, doi:10.3390/jcm10214832_

Round 1

Reviewer 1 Report

I have read the article about CNI role in lupus nephritis treatment and have some comments to the author:

  • when you describe the mechanism of actions of CNI it would be great to have a picture illustrating that. It would be easier for the reader to follow the text.
  • the side effect part could be put in the table, in the shortened version. You could elaborate on nephrotoxicity separately by using this biopsy picture. Or you could improve the text flow in this part.
  • the future perspectives should be also discussed.
  •  
  •  

Reviewer 2 Report

In this review the mechanism of calcineurin inhibitors especially in lupus nephritis is clearly presented in this very well written article. The authors clearly pointed out and give an overview of the side effects of CNI`s, the mechanism of CNI`s and they sum up different treatment combinations in lupus nephritis. I just would recommend minor adaptions. It should be pointed out that at present considering the KADIGO guideline CNIs is not first line therapy for lupus nephritis class III and IV. Table 3 is quite overloaded and should be more simplified and structured, if possible.  Even so around 90% suffering from SLE are women only in a short sentence the use of CNI`s in pregnancy is mentioned. More clarification on that point would be of great interest.

Reviewer 3 Report

Point 1

Line 48: Calcineurin modulates cellular response to 2nd messanger Ca2+. 

Ca2+ is not a second messanger, Ca2+ dependent calmodulin is. Please modify the sentence accordingly.

Line 59: Naïve T-cell requires two different signals to become activated. 

Naive T cells infact require 3 signals to activate, and the third signal being cytokines secreted by APC's to direct the T cell phenotype.

Line 151: Secretion of IL2 by Th1 activates B cells to secrete antibodies.

Can the author state a reference for this statement? It is known that IL4 polarized Th2 cells primarily act on B cells to initiate the humoral response.

Figure 1: Has the image been specifically captured for this review article? Or has it been used else where also?

Line 393: What is mg/kg/die?

Round 2

Reviewer 1 Report

Congratulation on this great review. You have properly addressed all my comments.